# Resilience Scale Psychometric Study. Adaptation to the Spanish Population in Nursing Students

**DOI:** 10.3390/ijerph17124602

**Published:** 2020-06-26

**Authors:** Ana M. Tur Porcar, Noemí Cuartero Monteagudo, Vicente Gea-Caballero, Raúl Juárez-Vela

**Affiliations:** 1Faculty of Psychology, University of Valencia, Avda. Blasco Ibáñez 21, 46010 Valencia, Spain; 2Pediatric Nurse Lecture, Catholic University of Valencia, C/Espartero 7, 46007 Valencia, Spain; 3Nursing School La Fe, Adscript Center of University of Valencia, 46026 Valencia, Spain; gea_vic@gva.es; 4Research Group GREIACC, Research Health Institute La Fe, Pabellon Docente. Torre H. Avda. de Fernando Abril Martorell 106, 46026 Valencia, Spain; 5Department of Nursing, University of La Rioja, C/Duquesa de la Victoria, 88, 26003 Logroño, Spain; raul.juarez@unirioja.es; 6Research Group BMP, Idi-Paz, Hospital La Paz, Paseo Castellana 261, 28046 Madrid, Spain

**Keywords:** resilience, scale validation, psychometric properties, nursing students, coping

## Abstract

Nursing students and professionals are exposed to highly stressful clinical situations. However, when confronted with stress, which is exacerbated by academic and professional situations, there is a great disparity between those who do not know how to respond suitably to the demands from patients or teachers due to a lack of competence and personal resistance, and those who are more resilient and develop a greater range of strengths. This research aims to analyse the validity and psychometric characteristics of a questionnaire on resilience adapted to Spanish nursing bachelor’s degree students. The participants were 434 undergraduate nursing students from the province of Valencia (Spain) between 17 and 54 years of age (Mean, M = 21; Standard Deviation, SD = 0.320), 104 of whom were men (24%) and 330 women (76%). A cross-sectional group evaluation was carried out in the university itself, adhering to the ethical standards of the Declaration of Helsinki. Based on the descriptive, factorial, exploratory and confirmatory analyses, it was possible to confirm the suitability of the questionnaire and its adaptation to nursing students. The model is thus suitable for evaluating the population under study. Furthermore, there are statistically significant differences depending on age and gender. The results show that the questionnaire analysed is suited to evaluating resilience among Spanish nursing students, thereby justifying the adaptation of a scale of this nature to foster resilience among nursing students and nurses in professional life, who are exposed to critical situations with patients’ suffering, deterioration or death. Our study highlights important practical implications: Spanish nursing studies involve theory and practice, but students and nurses in professional life have to confront critical situations of patients’ suffering, deterioration, or death. These situations cause stress and feelings of impotence that may lead to chronic stress and even suicidal thoughts.

## 1. Introduction

Recent research has demonstrated the need to boost resilience among nursing students and professional nurses to improve strategies to confront situations of patients’ suffering [1]. In general, nursing students are subject to scenarios that involve stress due to the demands of academic matters as well as due to work experience in clinical environments. In such environments, they are exposed to situations with patients’ suffering, illness or death, and they may feel afraid of failure [2,3].

Fears and self-destructive ideas are generally widespread among nursing students, since they do not know how to respond adequately to the patients’ demands due to a lack of competence and personal strength required to confront the situation itself. In this vein, it should be remembered that suicide rates among the Spanish populace from 15 to 39 years of age have been confirmed to have risen to 3.94% [4]. These indices may even reach 14% among nursing students [2].

On the other hand, it has been seen that the development of a more resilient nursing workforce has been identified as a strategy to improve the individual’s response to workplace adversity, which has positive implications for staff wellbeing and delivery of person-centred care [5]. Resilience also helps reduce the stress aggravated by academic and professional situations [6]. This is why there is interest in the challenge faced by nursing teachers as regards the need to train their students in resilience [1,7].

However, in developing intervention strategies there is currently a paucity of simple, easily applicable questionnaires to evaluate nursing students’ resilience. This study’s aim was therefore to analyse the psychometric properties of the Resilience Appraisals Scale (RAS) questionnaire [8] among Spanish nursing bachelor’s degree students. In a recent review, it was observed than most of the studies are for English-speaking countries, so the results of this work may help understand the study of resilience in a non-English-speaking society [9].

### 1.1. Background

Psychological resilience refers to the human capacity to adapt to trauma, adversity, difficulties and stressful factors in daily life [10,11]. It is related to the organization of mental processes and behavior geared towards activating thoughts for personal protection in the face of the possible negative effects of stressful factors and the demands of work [12].

Worldwide there is a need for nurses while at the same time a worrying number of nursing students struggle to complete their education [13].

There is a need for approaches that not only educate an adequate number of nurses but also equip them with the skills to manage the complex challenges inherent in daily nursing practice. In this context, resilience seems to be an important asset [14].

In clinical and academic environment, nursing students have the perception of little skill in the clinical practice increasing stress. Fears and self-destructive ideas may arise either due to these environments or because they do not have an adequate response to the situation [15]. It has been confirmed that in nursing, the skills for interacting with colleagues and to manage the complex challenges inherent in daily nursing practice, patients and their relatives are just as important as clinical knowledge [16]. Therefore, developing skills, such as resilience, as part of nursing programs allows students to be better prepared to deal with the unique challenges in nursing practice [17].

Developing skills for social interaction, a person’s internal attributes are involved as well as external factors such as support from their social environment like their family, friends and work colleagues [18]. That is why there is an interest in developing processes to handle other people’s negative emotions as well as one’s own [19], such as positive self-appraisals [8] and deeper activity at work that leads to job satisfaction as well as satisfaction for the patient [20]. “Deep activity” means knowing how to express emotions suited to a specific situation [6].

Resilience, understood to mean a process, involves the capacity for positive adaptation in the face of adversity, with dynamic aspects that can be adjusted to the situations in continuous evolution and adaptation [21]. These mental processes interact in the social context of adversity and strengthen those who use them [22].

Both individual and contextual factors have an effect on the capacity to develop resilient factors, which involve processes of regulation and active coping in confronting adversity [9,23]. The more resilience an individual reports, the more likely she is to apply active coping to mitigate negative health outcomes. Moreover, resilience acts as a mediator that decreases adverse impacts on health outcomes [24]. Active coping strategies, then, are associated with psychological resilience and involve efficient ways of confronting the situation. They are one of the essential components of psychological resilience, given that active coping helps in adopting strategies to mitigate the effects of adversity and to contribute to psychological well-being [23]. People who tackle the problem by confronting it and seeking a solution that focuses on the problem steadily change the way they see the problem and reduce their stress. Significant differences have been found in the way people confront stressful situations. For example, in general, women are more likely to use strategies that focus on emotion and to seek social support [25].

Resilience is also associated with positive emotions, which are more common in resilient individuals and act as support in recovering from stressful processes [26] Furthermore, on comparing students in their first and last year of the bachelor’s course in nursing, it seems that the capacity for resilience increases with age [27].

In recent years, resilience has been a subject of study among nurses, and it has been shown that it may have a positive impact on the future of people who wish to pursue this profession [28]. On the whole, the different reviews address factors related to resilience such as emotional intelligence, detachment, gender and conflicts among colleagues [28,29].

### 1.2. Aims

The main aim of this research is to analyze the questionnaire’s psychometric characteristics and the validity of the Resilience Appraisals Scale construct adapted to Spanish nursing bachelor’s degree students [8]. This questionnaire has already been applied to different populations such as people at risk of suicide, the elderly and young people [30]. It would thus seem to be an ideal measure to apply to nursing staff taking into account the three factors it measures: resilience related to social support, situation coping and emotion coping [8].

## 2. Materials and Methods

### 2.1. Structure and Participants

There were 434 nursing bachelor’s degree students from the province of Valencia involved (74% of the students enrolled in all of the nursing faculties in Valencia province). The city of Valencia, in the east of Spain, is the third biggest provincial capital in the country in terms of population and has five nursing faculties, whose students attend from all over the Valencia region and other parts of Spain. The participants in this investigation were between 17 and 54 years of age (Mean, M = 21; Standard Deviation, SD = 0.320). There were 104 men (24%) and 330 women (76%). Their marital status was as follows: single (71.6%), married (3.7%), divorced (1.4%) or common-law partner (23.3%). There were 95.2% with no children. Most were Spanish (94%), while the other 6% came from other European Union countries (3%), Latin America (2%) and Asia (1%). There were 80.6% with less than one year’s work experience or none at all, while 12% had up to five years’ work experience and 7.4% had over five years’ work experience. Only 31.6% of the students with work experience had experience in attending to patients, however. There were 81% studying nursing as their vocation. There was not a sample size, because the questionary was administered to the entire universe of the study.

The inclusion criteria were as follows: (i) they should be studying the first year of nursing and (ii) they must give their consent to take part. There were no exclusion criteria stipulated. The proportion of men to women participants in the research is in keeping with the proportion of men and women that study nursing in the province of Valencia (Spain), where men account for about 20% [31].

### 2.2. Data Collection

Before carrying out the evaluation, permission was obtained from the institutions then from the students studying there themselves. This was accompanied by the informed consent forms. Participation was voluntary. The anonymity and confidentiality of the information were maintained. They could leave the investigation at any time, though no students in fact left. The principles set out in the Helsinki Declaration for research with human subjects were observed. The evaluation used a cross-sectional group study in the university itself, with the students receiving the instructions and answering them individually. The questionnaire was disseminated by a professor during the class, the professor was responsible for the administration and the custody.

The items were translated into Spanish with the technical help of a bilingual professional born and educated in Great Britain with good knowledge of Spanish. It was preceded by a pilot study with university students. The authors of the questionnaire have been informed about the study.

### 2.3. Questionnaires

The RAS questionnaire (Resilience Appraisal Scale) [8] has 12 items. It uses a Lickert scale with five possible answers (1 = completely disagree; 5 = fully agree). The original scale takes into account three factors (social support, emotion coping and situation coping) as factors that help to confront stressful situations and to protect from negative thoughts that may lead to suicidal thoughts [8]. In the original study, Cronbach’s Alpha = 0.88 for the global scale, 0.93 for the social support sub-scale, 0.92 for emotion coping and 0.92 for situation coping [8]. In our study, Cronbach’s Alpha is as follows: the global scale, *α = 0.83*; social support, *α = 0.76*; emotion coping, *α* = 0.91 and situation coping, *α* = 0.84. Cronbach’s Alphas above 0.70 may be considered acceptable [32].

### 2.4. Data Analysis

An internal consistency analysis was carried out for the instrument via Cronbach’s Alpha, descriptive analyses and exploratory and confirmatory factorial analyses, using the statistical software SPSS 24.0 and AMOS 16.0 (1995, Chicago, IL, USA). The Promax oblique rotation method was used by means of analysing the principal components [33] to precisely reflect the interaction between the elements [34].

For the model’s degree of fit, the Kaiser–Meyer–Olkin (KMO) measurement was checked, which analyses scores for each of the variables and their predictability based on the others (range 0 to 1) and Bartlett’s test of sphericity to see that the correlation matrix is not an identity matrix. The result of χ^2^ divided by the degrees of freedom indicates that values of lower than 5 are indicators of a good fit [35]. The Root Mean Standard Error (RMSEA) was also obtained as well as the Standardized Root Mean Square Residual (SRMR), which must be below 0.08 [36] and the Goodness of Fit Index (GFI), Adjusted Goodness of Fit Index (AGFI) and the Tucker–Lewis Index (TLI), which should preferably be above 0.90 [36].

Student’s t test was used to observe possible significant differences in the resilience factors between the men and women in the sample and between the youngest and oldest people. The separation point was 24 years of age.

## 3. Results

The results from the descriptive statistics indicate that the population analysed obtained above average global mean scores in all items but especially in items 1, 2, 6 and 10. These items correspond to the social support factorial analysis group with a degree of dispersion lower than one (Table 1). The items showing greater dispersion are 3 and 4, which say “My family or friends are very supportive of me” and “In difficult situations, I can manage my emotions”.

The scree plot shows a tri-factorial structure (Cattell’s plot). The Kaiser–Meyer–Olkin measure for sampling adequacy (KMO = 0.855) and Bartlett’s sphericity test (chi-squared = 2617.20; df = 66; *p* < 0.000) indicate the factorial adequacy and the interdependence between the items. The explained variance is 69.07%. The factorial weights range from 0.656 to 0.908. Factor 1 refers to emotion coping, Factor 2 to situation coping and Factor 3 to social support (Table 2).

Afterwards, a correctional analysis was carried out among the items, grouped according to the factors obtained in the factorial structure. Table 3 shows the results of the convergent validity among the items, distributed according to the factors obtained in the EFA. As one can see, the relationships are quite high, with indices above 0.500 (*p* = 0.001) except between item 6 and item 1 and between item 6 and item 2, which despite being a little lower are also near 0.400 (*p* = 0.001). These results corroborate the connection between the factors’ items (Table 3).

### 3.1. Confirmatory Factorial Analysis (CFA)

The CFA gave good adjustment indices for the model (Table 4). Chi-squared divided among degrees of freedom is below five. The RMSEA and SRMR indices are below 0.08, and the other indices (GFI, AGFI and TLI) are above 0.90 [36] (Table 4 and Figure 1). Thus, the model is suitable for evaluating nursing students in the Spanish population.

### 3.2. Differences by Participants’ Sex and Age

Having seen the results and the adequacy of the model, it was then of interest to observe if there are significant differences in the resilience factors according to the participants’ sex and age. Sex was categorised as a dummy variable, assigning 1 for men and 0 for women. The age of the population was split into two groups. The first group was made up of students under 24 years of age, and the second group included those 24 years of age or over.

By sex, the population shows significant differences in emotional coping, situation coping and social support. The results of the t-test indicate that the women have better social support indices than men (women: M = 4.34, SD = 0.642; men: M = 4.17, SD = 0.532; *t* = −2.367; sig = 0.018) but lower indices in emotional coping (women: M = 3.16, SD = 0.879; men: M = 3.76, SD = 0.698; *t* = 3.862; sig = 0.0001) and situation coping (women: M = 3.69, SD = 0.586; men: M = 3.94, SD = 0.558; *t* = 4.235; sig = 0.0001) (Figure 2).

Age was split into two sub-samples resulting from the age separation cut-off point in the descriptive analyses. The first group was made up of students younger than 24 years of age, while the second group included those who were 24 or over.

The results giving the differences in means obtained via Student’s t test only show significant differences in situation coping, where the older students obtained higher indices (up to 24 years: M = 3.72, SD = 0.578; 25 and over: M = 3.93, SD = 0.7634; *t* = −2.390; sig = 0.017). Although no significant differences appear in emotion coping or in social support, it is seen that the younger students’ emotion coping indices tend to be lower (up to 24 years of age: M = 3.29, SD = 0.889; 25 or over: M = 3.40, SD = 0.780; *t* = −0.870; sig = 0.385) while their social support indices are higher (up to 24: M = 4.31; SD = 0.623; 25 or over: M = 4.18; SD = 0.780; *t* = 1.396; sig = 0.163). The mean scores appear in Figure 3.

## 4. Discussion

The aim of this article is to analyse the psychometric properties of a resilience questionnaire adapted to Spanish nursing students [8]. The procedure was based on a Spanish translation of the questionnaire with the support of a professional British native with excellent knowledge of Spanish. A pilot study was carried out beforehand with university students, obtaining good results. The items in English and Spanish appear in Table 2.

In recent years, the role played by resilience in nursing students and in professional nurses has steadily been demonstrated [1]. As a result, nursing teachers are recommended to prepare bachelor’s nursing degree students with strategies that may help them tackle complex situations with patients’ suffering or death, or their fear of a possible professional failure [37].

We thus consider it necessary to have evaluation instruments that may be used to find Spanish nursing students’ levels of resilience. Based on the descriptive, factorial, exploratory and confirmatory analyses, it was possible to confirm the suitability of the questionnaire and its adaptation to nursing students.

Internal consistency, using Cronbach’s Alpha, was over 0.80 for each of the items. In the analysis of items grouped by EFA factors, Cronbach’s Alpha was 0.91 in emotion coping (Factor 1), 0.84 in situation coping (Factor 2) and 0.76 for social support (Factor 3). 

Cattell’s plot confirmed a tri-factorial structure in agreement with the structure obtained by Johnson et al. [8]. The twelve resulting items are the same as those from the original scale, as is the distribution of the items in factors, obtained in the EFA (Table 2). The factorial weights are high, between 0.656 and 0.908, so very near to 1. A similar situation occurs in the correlation analysis among the items grouped according to the factors to which they belong. The correlations were quite high, as seen in Table 3. Lastly, the CFA’s goodness of fit indices confirm that the model is suitable for evaluating nursing students in the Spanish population. 

There follows an analysis of the possible differences among nursing students by sex and age. The results of the Student’s t test demonstrate that significant differences appear between men and women in the three resilience factors. The men obtained better indices in emotion coping and situation coping, whereas the women showed higher indices in social support. Although till now we have not seen research comparing male and female nursing students’ resilience, these results are in line with the preceding investigation where, faced with stressful situations, women were more likely to seek social support and focus on emotion coping, whereas there were hardly any differences between men and women in terms of coping focused on the situation [25]. 

Finally, on comparing the scores for the resilience factors in the population split into age ranges (up to 24 years or 24 and over), the results showed significant differences in situation coping, with higher indices among the older students. No differences appear in emotion coping or in social support, however. We can thus conclude that students of 24 years and over are more prepared to confront the demands of the situations by focusing on the problem (situation coping). These results back up the research by Pitt et al. [27] (2014), who concluded that resilience increases with age. In the research by Pitt et al. [27], however, resilience is considered to be a trait similar to emotional stability and is defined using concepts such as volatility and anxiety. This way of understanding resilience is not very widespread. In a recent review, they observed risks involved in studying resilience focussing on personal traits, given that it may lead to blaming the person who does not respond with resilience in the face of adversity [9].

## 5. Limitations

The limitations to this research include, firstly, its broad-based nature, which may cause biases related to the situation being evaluated because of the time and place of the evaluation. Another limitation arises because the results are based on self-appraisal reports, which may also contain biases involving a desire to look good, for social desirability or due to the emotional and physical state of the person filling in the questionnaire. A third limitation is linked to the sample being made up mostly of women (more than 75%). This limitation must thus be taken into account in the analysis by different sexes. Even so, this proportion is equivalent to the proportion of men and women students in nursing in general. In Spain, 80% of nursing students are women [31]. 

## 6. Conclusions

There is ever more research showing the need to foster resilience among nursing students [1,27,28]. Spanish nursing studies involve theory and practice, but students and nurses in professional life have to confront critical situations of patients’ suffering, deterioration or death. These situations cause stress and feelings of impotence that may lead to chronic stress and even suicidal thoughts [2]. In order to tackle these situations, the positive effects of resilience have been shown among nursing students and professionals. To this end, nursing teachers need simple, easily administered tools to help them evaluate their students’ resilience to find out what resources and mechanisms are most suitable to boost it [1,7,38].

The questionnaire that was analysed is simple and easy to administer and understand. It has demonstrated that it has been adapted for evaluating resilience in Spanish nursing students. In future research, it would be useful to evaluate nursing students from other societies and cultures in order to generalise the results and draw conclusions about the questionnaire’s validity on a larger scale. Furthermore, it could be interesting to compare these results to the ones obtained in other geographic contexts and cultures.

## Figures and Tables

**Figure 1 ijerph-17-04602-f001:**
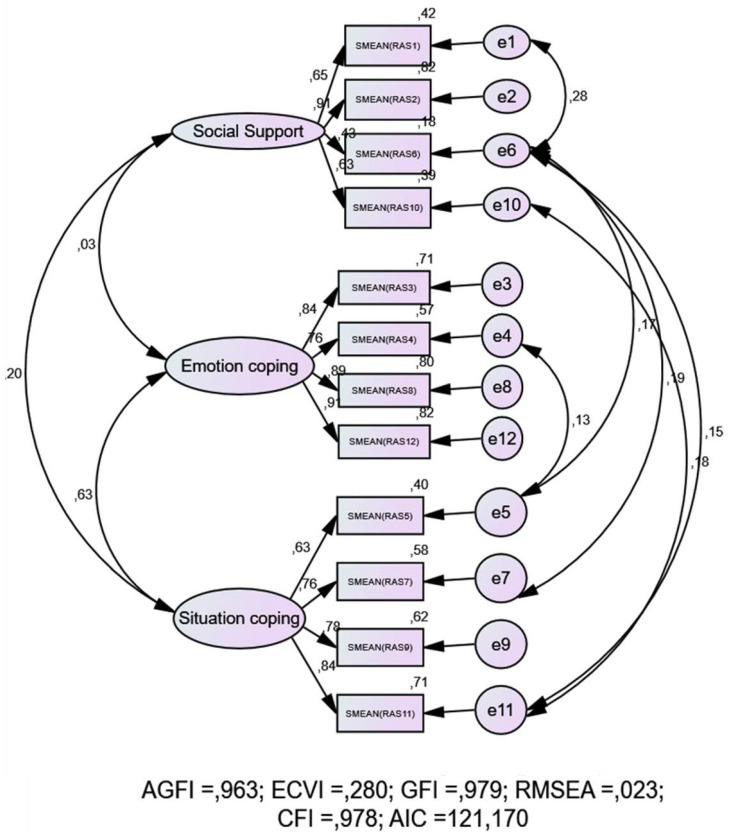
Results of Confirmatory Factorial Analysis (CFA) (Flow Chart). AGFI, adjusted goodness of fit index; ECVI, expected cross validation index; GFI, goodness of fit index; RMSEA, root-mean-square error of approximation; CFI, comparative fit index; AIC, Akaike information criterion.

**Figure 2 ijerph-17-04602-f002:**
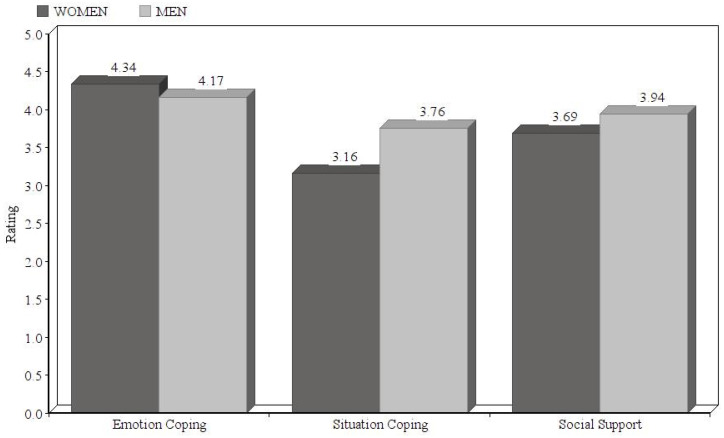
Differences between men and women in terms of resilience factors.

**Figure 3 ijerph-17-04602-f003:**
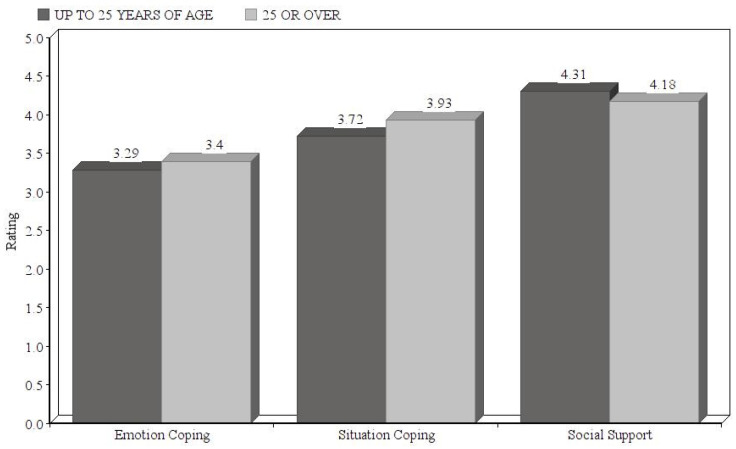
Differences in resilience factors by age.

**Table 1 ijerph-17-04602-t001:** Descriptive statistics and Cronbach’s Alpha for the items in the questionnaire.

Items	Mean	Standard Deviation	Reliability (Cronbach’s Alpha)
Item 1	4.29	0.87	0.836
Item 2	4.50	0.78	0.832
Item 3	3.27	1.01	0.808
Item 4	3.21	1.04	0.811
Item 5	3.71	0.72	0.816
Item 6	3.92	0.91	0.835
Item 7	3.71	0.70	0.813
Item 8	3.34	0.94	0.808
Item 9	3.80	0.71	0.813
Item 10	4.47	0.78	0.834
Item 11	3.76	0.70	0.808
Item 12	3.37	0.93	0.802

Exploratory factorial analysis (EFA).

**Table 2 ijerph-17-04602-t002:** Exploratory factorial analysis factorial weights.

Items	Factor 1	Factor 2	Factor 3
Item 1. If I were to have problems, I have people I could turn to.			0.819
Item 2. My family or friends are very supportive of me.			0.850
Item 3. In difficult situations, I can manage my emotions.	0.893		
Item 4. I can put up with my negative emotions.	0.839		
Item 5. When faced with a problem I can usually find a solution.		0.764	
Item 6. If I were in trouble, I know of others who would be able to help me.			0.656
Item 7. I can generally solve problems that occur.		0.843	
Item 8. I can control my emotions.	0.908		
Item 9. I can usually find a way of overcoming problems.		0.832	
Item 10. I could find family or friends who listen to me if I needed them to.			0.738
Item 11. If faced with a set-back, I could probably find a way round the problem.		0.855	
Item 12. I can handle my emotions.	0.908		

**Table 3 ijerph-17-04602-t003:** Correctional analysis among the items, grouped according to the exploratory factorial analysis.

**Factor 1**	**Item 3**	**Item 4**	**Item 8**	**Item 12**
Item 3	-			
Item 4	0.666 *	-		
Item 8	0.742 *	0.677 *	-	
Item 12	0.756 *	0.663 *	0.789 *	-
**Factor 2**	**Item 5**	**Item 7**	**Item 9**	**Item 11**
Item 5	-			
Item 7	0.523 *	-		
Item 9	0.502 *	0.591 *	-	
Item 11	0.519 *	0.647 *	0.667 *	-
**Factor 3**	**Item 1**	**Item 2**	**Item 6**	**Item 10**
Item 1	-			
Item 2	0.600 *	-		
Item 6	0.373 *	0.373 *	-	
Item 10	0.568 *	0.568 *	0.406 *	-

* sig ≤ 0.001.

**Table 4 ijerph-17-04602-t004:** Confirmatory factorial analysis adjustment indices.

χ^2^	DF	RMSEA	RMR	GFI	AGFI	TLI
55.170	45	0.023	0.014	0.979	0.963	0.978

DF, Degrees of freedom; RMSEA, Root-Mean-Square Error of Approximation; RMR, Root Mean Square residual; GFI, Goodness of Fit Index; AGFI, Adjusted Goodness of Fit Index; TLI, Tucker Lewis Index.

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
