# Peer review of "Resilience Scale Psychometric Study. Adaptation to the Spanish Population in Nursing Students"

_ijerph, 2020, doi:10.3390/ijerph17124602_

Round 1
Reviewer 1 Report
A very well written manuscript for an interesting topic.
Minor comments:
1-so a pilot study was conducted after the questionnaire was translated, other than the full study right?- but no details were provided on how many people piloted it and what were the results of the pilot study- what changes were made in the questionnaire after the pilot study?
2-Why was a sample size for participants not calculated?
3- How was the questionnaire disseminated? was it a paper questionnaire or electronic? If it was a paper questionnaire who gave it out and when?
4-why were the differences not explored in relation to the students' work experience?
5-Graphs' axes need to be labelled appropriately.
Author Response
Reviewer 1.
1-So a pilot study was conducted after the questionnaire was translated, other than the full study right?- but no details were provided on how many people piloted it and what were the results of the pilot study- what changes were made in the questionnaire after the pilot study?
Thanks so much for this comment. We performed a study to confirm the suitability of the questionnaire and its adaptation to nursing students. The items were translated into Spanish with the technical help of a bilingual professional born and educated in Great Britain, with good knowledge of Spanish.
We performed a back-translation. The translation and back translation is one method widely used in health sciences with 3 steps: in the first step, we translate into Spanish the scale, in the second step we completed translation back into the original language, after that we sharing that new translation with the original text; in the last step reconciling any meaningful differences between the two.
The purpose of our study was to describe based on the descriptive, factorial, exploratory, and confirmatory analyzes of how the scale fitted in nursing students. We performed the piloted study about the 434 nursing bachelor’s degree students from the province of Valencia. We used a tri-factorial structure (Cattell’s plot). The Kaiser-Meyer-Olkin measure for sampling adequacy (KMO = 0.855) and Bartlett's sphericity test (chi-squared = 2617.20; df = 66; p <0.000) indicate the factorial adequacy and the interdependence between the item who allowed us to performed the confirmatory analysis afterwards, to fulfill the objective of the research. With the results we described was not necessary any adjustment in the questionary ( good fit) and we could continue with the confirmatory analysis.
2-Why was a sample size for participants not calculated?
Thanks you for this question. We did not sample size because it was administered to the entire universe of the study. We add this information to the paper. Thanks.
- How was the questionnaire disseminated? was it a paper questionnaire or electronic? If it was a paper questionnaire who gave it out and when?
Thanks for this question. The questionnaire was disseminated by a professor during the class , the professor was the responsible of the administration. We add it to the text thank you very much.
4-why were the differences not explored in relation to the students' work experience?
Thanks for this question. We considered future investigations could be carry out to explore the relation to the student´s work experience. In this paper we focused mostly in psychometric characteristics. We will do in a future. Thanks so much.
5-Graphs' axes need to be labelled appropriately.
We are very sorry. We fix it, thanks so much
Reviewer 2 Report
I have reviewed the paper: Resilience scale psychometric study. Adaptation to the Spanish
population in nursing students
Manuscript ID: ijerph-828723
Key words
Ok
Abstract
The abstract does not indicate the impact of this work. That is, what will it be useful for or of practical application?
Furthermore, authors should review carefully written English throughout the manuscript. They use expressions and words that are not appropriate.
Introduction
Line 28
add comma after environtment,
Line 32-35 improving the English writting
Background
Line 53-55 improving the English writting
In the background section, the authors do not make an in-depth review of resilience in the context of nursing students.
The authors make a detailed description of the concept of resilience, but do not focus the subject in the academic context.
In some paragraph they introduce some sentences that refer to the students, but without making an argument in this context.
There is an extensive literature on resilience, resilience, and nursing; there are fewer references to resilience and nursing students, but there is also.
Authors should rewrite this section, directly focusing on the topic of resilience, a brief definition and description of resilience papers in university nursing students
In this description of literature on the topic, they must point out the research hole, to explain the objective with which it intends to fill this gap
Aim
0k
Methods
Data collection
In the Data collection section, the authors must describe how the data was collected, by what tool, how the data was stored
Results
Review and improve table 2, 3 and 4
Line 196 sig= HERE, IT´S SPACE 0.0001)
Improve Graph 2. Differences in resilience factors by age.
Discussion
ok
Conclusion
ok
References
Line 295 remove a period at the end of the sentence
references should be thoroughly examined.
There are several errors, it does not have the same format, margins, etc.
Author Response
The abstract does not indicate the impact of this work. That is, what will it be useful for or of practical application?
We are very sorry. We add to the abstract. Thanks so much.
Furthermore, authors should review carefully written English throughout the manuscript. They use expressions and words that are not appropriate.
We are very sorry. English is always a challenge for non-native English-Speaking. We highlighted some expressions changed by the native English enterprise where we sent the paper for translation. Thanks so much
Introduction, Line 28 add comma after environment.
We are very sorry. We fix in the text. Thanks so much
Line 32-35 improving the English writing
We are very sorry. We fix in the text. Thanks so much
Background. Line 53-55 improving the English writing
We are very sorry. We fix in the text. Thanks so much
In the background section, the authors do not make an in-depth review of resilience in the context of nursing students.The authors make a detailed description of the concept of resilience, but do not focus the subject in the academic context.
We are very sorry. We fix in the text. There are not so many students how to describe a depth review of resilience into the Spanish population. We add some comments and some references in addition.
Thanks so much for this comment.
In some paragraph they introduce some sentences that refer to the students, but without making an argument in this context.
We are very sorry. We fix in the text. We have tried to follow the common thread about fears, stress and anxiety in nursing students. We reissued it. Thank you
There is an extensive literature on resilience, resilience, and nursing; there are fewer references to resilience and nursing students.
We are very sorry. You are right. We have reissued it. Our research focuses on Spanish nursing students, and we have not identified researchers carry out in this field of research. Therefore, we had to try identify the common characteristics.
Reviewer 3 Report
First, thank the researchers for submitting the work to the journal. The authors carry out a validation of a resilience scale adapted to the Spanish population. The analysis carried has some things that need to be improved to be considered for publication. - Cronbach alpfa statistic is not sufficient to measure the validity of a questionnaire. Although this statistic has been widely used in social research, we complemented it with other analyses to avoid biases inherent to the test (Dunn, Baguley & Brunsden, 2013; Sijtsma, 2008), these being composite reliability (CR) and average variance extracted (AVE) indices.- Add in table 4 a literature review good indices to compare. - In table 1: TAlpha value cannot be calculated for a single item, it must be calculated by dimensions or group of items. What the researchers present will not be the alpha if the element is removed? - Its necessary present figure with the final model in AMOS with values. - How was the information from the questionnaire collected?
Author Response
In Data collection section, the authors must describe how the data was collected, by what tool, how the data was stored
We deeply appreciate this comment. You are absolutely right. We have introduced it with the comments of reviewer 1.
Thank you.
Results. Review and improve table 2, 3 and 4
Thanks so much for this comment. We fix it.
Line 196. sig= HERE, IT´S SPACE 0.0001)
Thanks so much for this comment. We fix it.
Improve Graph 2. Differences in resilience factors by age.
Thanks so much for this comment. We fix it, We are changed all graphics.
References
Line 295 remove a period at the end of the sentence
references should be thoroughly examined
Thanks so much for this comment. We fix it. All the reference has been update. In addition, format, and margins has been checked. Thanks so much, we are very sorry.
T-Alpha value cannot be calculated for a single item, it must be calculated by dimensions or group of items.
The values have been included in graph 1, which we have added, adding a new graph of the confirmatory analysis with the relevant values.
How was the information from the questionnaire collected?
Thanks so much for this comment. We fix it with the comments of the first reviewer.
Amos Graphic
Thanks so much for this comment. We fix it.
Note to the Reviewer 3.
We greatly appreciate his comments, indeed, Cronbach's alpha statistic is not enough to measure the validity of a questionnaire, we are agree with you, but in the statistical analysis, the exploratory factor analysis should also be considered as key in the psychometric properties of the questionnaire. A combination of the exploratory factor analysis and the confirmatory, once established with Cronbach's alpha, gives a good fit to the questionary. In any case, we are at your disposal for as many changes as you deem appropriate and we appreciate the opportunity to publish in your magazine. Thanks so much
Round 2
Reviewer 2 Report
I have reviewed the paper: Resilience scale psychometric study. Adaptation to the Spanish
population in nursing students
Manuscript ID: ijerph-828723
Table 1 table 1 is cut
Improve table 2.
I think it is better to put it only in English.
If there is a special interest, I suggest adding the translation.
Table 3 y 4 are cut
Graph 1 has no image quality. You must improve it a lot
Idem con grafph 2 and 3.
Authors should carefully review the bibliography again
I think that the authors have weaknesses in terms of the presentation of tables, figures and graphs.
This should not be an issue to delay publication of the paper. However, it is.
I would ask the authors to seek help to solve this question, which is a basic question in a scientific paper.
Author Response
Table 1 table 1 is cut
Improve table 2.I think it is better to put it only in English.If there is a special interest, I suggest adding the translation.
Thanks for the comment, we have observed that this is the case and have proceeded to modify it, adjusting it to the page margins.
Improve table 2.I think it is better to put it only in English.If there is a special interest, I suggest adding the translation.
Thanks for the comment, we have modified table 2. We have eliminated the Spanish translation, We have modified the size of the columns. We think it's more compact and clearer.
Table 3 y 4 are cut. If there is a special interest, I suggest adding the translation.We have reviewed these two tables, and in our document, both appear correctly. We don't know if it could be some kind of difference in the word processor version, but both tables are fine in our .txt.
Graph 1 has no image quality. You must improve it a lot. Idem con grafph 2 and 3.
Thanks a lot. We have changed Grapf 1, inserting the original. Now it looks good and doesn't pixelate.
Authors should carefully review the bibliography again
Thank you. We agree with your opinion. We have modified the graphics, increasing the size of the letters. Now you can read everything more clearly, and without pixels.
I think that the authors have weaknesses in terms of the presentation of tables, figures and graphs.
Thank you, we have tried to follow your recommendations so that the result is an article with better graphic presentation.
This should not be an issue to delay publication of the paper. However, it is.
I would ask the authors to seek help to solve this question, which is a basic question in a scientific paper.
Thank you, we hope that this final version will be adequate.
Reviewer 3 Report
Graphics must be improve, all graphics are pixelated
Author Response
Graphics must be improve, all graphics are pixelated
Thank you very much for the comment. We have modified all the graphics to improve them and make them suitable in quality.
We believe that in this version they are correct.